# {Not}ation: The In/Visible Visual Cultures of Musical Legibility in the English Renaissance

Eleanor Chan 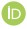

Department of Music, University of Manchester, Manchester M13 9PL, UK; eleanor.chan@manchester.ac.uk

**Abstract:** Legibility can seem as similar to the quintessence of musical notation, without which any attempt at musical inscription has fundamentally no purpose. Nevertheless, the visual culture of the English Renaissance is full of surviving examples that feature music books that are, fundamentally, illegible. Such instances are not useless, but rather shed vital light on the concerns of the visual culture of the English Renaissance, as well as what representation meant to the people who originally created and viewed these objects. What does it mean to include sheet music that merely looks similar to, but does not manifest, as legible notation? When does an object lose its semantic value? When do writing, notation, and signification pull lose from their seams and cease to be meaningful? Through the lens of a trio of objects (*Four Children Making Music* by the Master of the Countess of Warwick, an anonymous furnishing panel from Hardwick Hall, and a wall painting from High Street, Thame) that feature partially, or tantalizingly, legible musical notation, this paper seeks to explore the ramifications of visually depicting things that are and are not readable. Such objects have a graphic eloquence beyond the simple equation of sign and signified. Ultimately, entertaining the concept of illegible music notation within visual art objects as a deliberate stylistic choice, I argue that we can greatly enhance our understanding of what the notes on the page could mean in the English Renaissance.

**Keywords:** music notation; materiality of music; visual cultures of music; legibility; representation

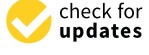



A quartet of children stand, gazing out of the frame at the viewer. A virginals keyboard slashes across the lower left side, skewed to an oblique diagonal angle in order to allow our eyes to appreciate the repetitive pattern of its keys and the girl's hands as they linger over them. This is the painting now known as *Four Children Making Music* by the Master of the Countess of Warwick, c. 1565 (Figure 1). At the heart of the painting, held loosely in the hands of the eldest boy, is the object that has gained most critical attention in recent years: a partbook of entirely legible musical notation, once believed to be a piece by William Byrd, but now identified by Kerry McCarthy as Domine ne in furore/Turbatus Est by none other than Josquin des Prez, the great behemoth of late fifteenth-century polyphony (McCarthy 2014). The text of the music is rendered in merely a few scribbles, enhancing what we would now call the photographic quality of the representation of the music notation, down to the bass clef and mensuration sign. In its shadow, the next eldest boy clutches a book filled with nothing but scrawls and shapes: at a distance, what appears to be musical notation is presented, but close up, it resolves itself to nothing but an enigmatic and suggestive imitation (Figure 2). The presence of the legible partbook, filled with music that can with relative certainty be ascribed to Josquin, has rightfully been interpreted as a prime insight into the musical and visual cultures of the mid-sixteenth century.[1] However, to date, the illegible partbook has not received such critical courtesy, though its implications are likely as fascinating and as illuminating as its more lauded partner. The value of musical objects such as this partbook does not begin and end with the question of legibility. Illegibility, or insouciant pretensions to legibility, can offer far more information (both to its original viewers/users and to historians of musical culture today) than simply that an artist or artisan failed to capture what something looked like. Musical scores in visual art objects are instances of 'multilayered materiality',[2] but they are

also traces of an intellectual culture typically interpreted as fundamentally invisible and immaterial. In such light, what does it mean to include sheet music that merely looks like, but does not manifest, as legible notation? When does an object lose its semantic value? When do writing, notation, and signification pull lose from their structural seams and cease to be meaningful? Through the lens of a trio of objects that feature partially, or tantalizingly, legible musical notation, this article seeks to explore the ramifications of visually depicting things that are and are not readable. Such objects, I argue, have a graphic eloquence beyond the simple equation of sign and signified.

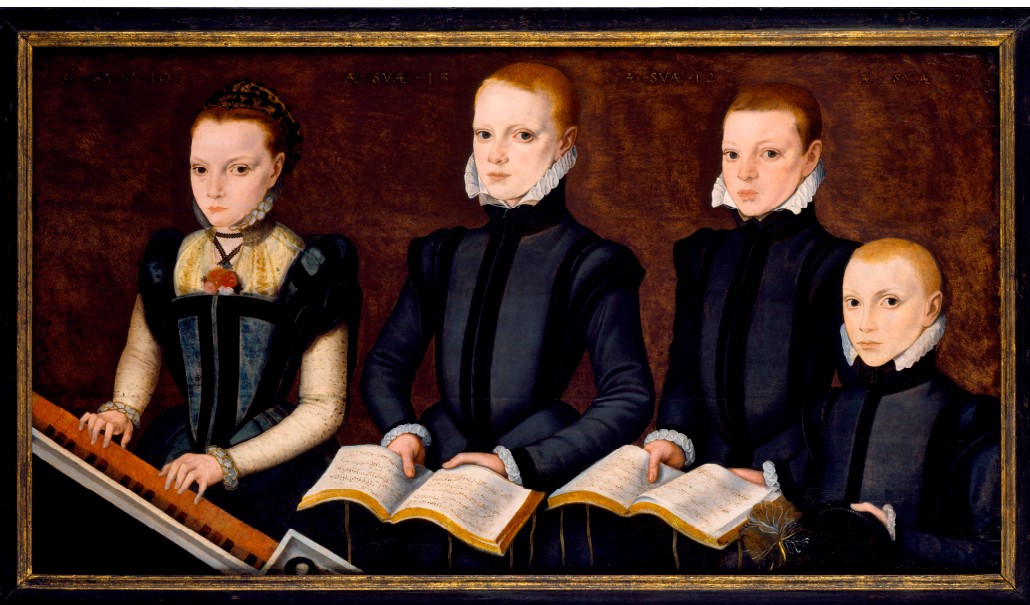

**Figure 1.** Master of the Countess of Warwick, Four Children Making Music, oil on panel, c. 1565 (Private Collection: by kind permission of the Weiss Gallery).

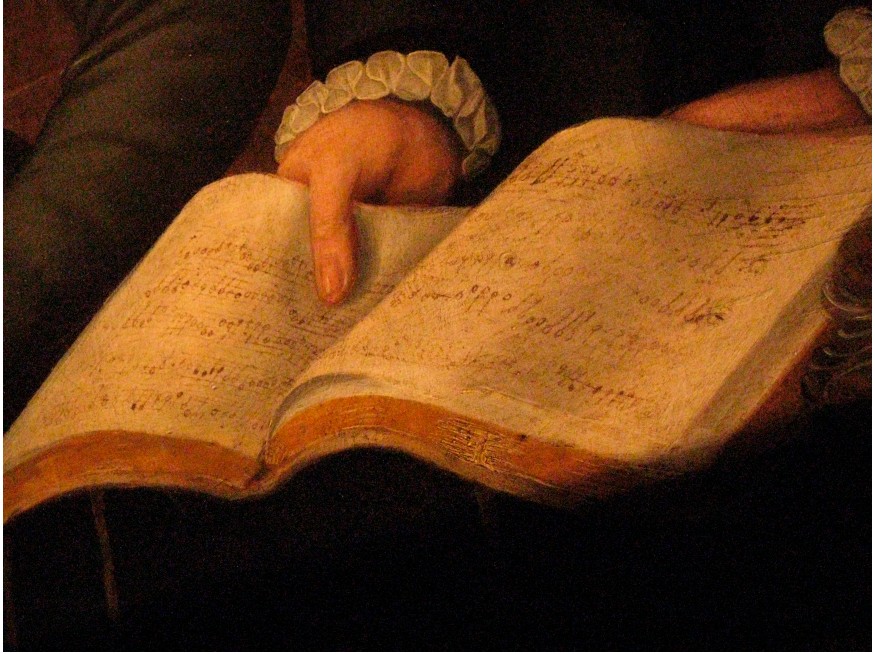

**Figure 2.** Detail of illegible partbook from Four Children Making Music (photograph credit: Benjamin Hebbert).

Musical notation is, of course, not the only form of graphic signification prone to slippage from meaning into decorative surface pattern and vice versa, in this period. It is worth briefly diverting from the matter at hand in order to anchor the musical within its broader contemporary context. Juliet Fleming has long since argued for the reconceptualization of writing in the English Renaissance, and a recognition that part of the appeal of printers' flowers and type ornament in this period is precisely because it plays into the slippage between the semantic and the asemantic. Such a fascination was driven not merely by increasing literacy. By the mid-sixteenth century, the English Reformation had percolated down throughout English society, such that the Bible became the cornerstone for Protestant piety.[3] The primacy of the Bible was infused into the visual culture of the period, resulting not in iconophobia, but, rather, in creative approaches towards what could be utilised for its visual appeal. One distinctive impact was the role that the visual appearance of printed text played in attitudes towards interior decoration and beyond. Once a musician at Henry VIII's and later Edward VI's court, Thomas Tusser (c.1524–1580) turned his attention to publishing; his immensely popular *Five Hundred Pointes of Good Husbandrie* (1573), for example, listed devotional and moralising 'poesies' or poems to incorporate into wall paintings in the 'hall', 'parler', 'gests chamber', and 'thine owne chamber'; *Five Hundred Points of Good Husbandrie* went into many editions throughout the Elizabethan period alone (Tusser 1573). These were the Elizabethan predecessors to the ubiquitous twenty-first century 'Live, Laugh, Love', as we can see from a brief cross-sample:

The wise wil spende, or geve & lend, yet kepe or have in store:

If fooles may have from hand to mouth, they pas uppon no more.

As hatred is the serpents noisome rod,

So friendship is the loving gifte of God.

The drunken frende is friendship very evill,

The frantike frend, is friendship for the Devill.

The quiet frend all one in word and dede,

Greate comfort is like redy golde at neede.

Some make the Chimney chamber pot, to smel like filthy sinke,

Yet who so bolde so sone to say, fough how these howses stinke (Ibid., fol. 40v-41r.).

Text was a key tactic in establishing a domestic space as inhabited, well kept, and fashionable.

Another was the representational convention of depicting depth in a synoptically planar manner than emphasized texture and multiple viewing perspectives, rather than one from a fixed point. English Renaissance visual culture was not, by and large, concerned with Florentine models of realism, but rather, in the capturing of the quality of a sumptuous drape of fabric, the grainy stitch-like textures of surfaces, and a narrative model governed by consciousness of this tactile ambience.[4] This focus on surface was coupled with compositional tactics that skewed various compositional elements to enable the viewer to read a given visual object without being bound to a fixed direction, as well as a preference for decorative art forms, rather than the 'fine' art forms, of panel painting and sculpture. Even in more recognizably so-called 'fine' art manifestations, such as the limned portrait miniature that the English Renaissance became famous for, planarity was important. Recent scholarship has debunked the idea that the art of limning was characterized by linearity in the sense of focusing on providing outline rather than fine modelling or shading.[5] Conservation has revealed that the paints that captured shading and modelling have corroded faster than the other pigments, and so the outlines of the miniatures of Teerlinc, Hilliard, and Oliver would have been far less stark. However, the quality of line was certainly a key interest in English visual culture. Strapwork patterns (so named for their resemblance to cut-offs or 'straps' from the leatherworking industry, which coiled into evocative spirals

and knots where they fell) crept across furnishings hard and soft, clothing, the pages of books, and jewellery. In the margins of books, they often constituted the printers' flowers that, as we saw above, Fleming argues, destabilized the boundaries between the semantic and asemantic properties of writing. Nowhere was this dynamic more potent than in the portrait miniature, which had arisen out of the technique of manuscript illumination, and which featured minutely figured undulating and stippled lines to capture their subjects' hair, the embroidery on their clothing, and the flow of drapery: a fine handling of silverpoint allied word with image and confected an evocative hinterland in between of text and not text. The dynamic interplay of semantic and asemantic drove much of the visual culture of the period.

Music, however, was a special case because of what it took to read it. At the beginning of the sixteenth century, the process of reading was conducted through solmization: a musical reader determined which hexachord (roughly, a precursor to a modern scale) the notated music designated as 'solmized' through the syllables 'ut', 're', 'mi', 'fa', 'sol', and 'la'. This six-note solmization system existed alongside the seven-note concept 'A', 'B', 'C', and so on of the diatonic system, and the theory of the eight-note pitch class we now call octaves. A successful reading was dependent upon the musician's visual acuity: in other words, the ability to correctly interpret the pattern that would enable performance. Although this was the process by which aspiring musicians, both amateur and professional, were taught to read whilst learning, it is likely that the ghost of its implications endured even as it became subconsciously ingrained in the act of performing. As a result, musical reading was a matter of inferring *through*, rather than *along*: musical notation was not a single unspooling thread, but rather a braided twisting of multiple threads, only one of which was explicitly visible or perceptible through the sense of hearing.

## 1. Notational Surfaces

Considered in this light, the illegible partbook to the right of *Four Children Making Music* takes on a different, subtler significance. Little is known about the circumstances of its creation; the identity of the sitters and of the painter or workshop is only patchily known. Roy Strong identified the 'Master of the Countess of Warwick' as the painter of a group of eight paintings, roughly in the style of Hans Eworth, and the artist was named after one of these works, the portrait of Anne Russell, Countess of Warwick (c.1569) (Strong 1969); since Strong's identification, more works have come to light, and there are now over fifty paintings associated with the Master and dating between 1561–1570. A possible identity of the Master has been floated by Edward Town as Arnold Derickson, who was not only known to have been employed by Hans Eworth, but also an associate of the portrait painter John Bettes the Elder, whose death in 1563 coincides with the beginning of the Master's surviving output (Town 2020). However, no securely attributed works by Derickson survive, and, so, it is impossible to state with any certainty that he is indeed the mysterious Master. Kerry McCarthy notes the high volume of portraits of children in the Master's output, which might suggest that they were in fact female.[6] There is certainly no reason to assume that the Master was male; this was, after all, the period during which Levina Teerlinc was the highest paid limner at court, from 1546 until her death, and she was paid a salary of £40 per annum, notably higher than Hans Holbein.[7] *Four Children Making Music* and other outputs, such as *Four Gentlemen Playing Primero* and *William Brooke, Baron Cobham and his Family at Table* (c. 1567) also display a marked resemblance to the works of Anthonis Mor in the figuration of hands and face; Mor had been in England to paint a portrait of Mary I in 1553 and returned at some point in the late 1560s, so again it is possible that the Master had served as one of his studio assistants. In any case, and regardless of whether the identity of the Master remains forever unsolved, it is evident that they had considerable skill and were highly valued amongst the court.[8] As Town notes, 'stylistically, the Master of the Countess of Warwick seems to have been a major influence on George Gower, to the point where it is almost impossible to tell where the career of the former ends and the latter other begins [ . . . ] What has not been mentioned is that there is

a strong affinity between the work of Gower, the Master of the Countess of Warwick, and the early work of Nicholas Hilliard. All three artists privilege the depiction of linear detail drawn with the brush over any attempt to achieve an illusionistic sense of depth, with the stiff deportment of their sitters, often imparting a haughty demeanour'.[9]

Such crucial features and traces indicate that the choice to present an illegible partbook alongside a legible iteration was deliberate. The presence of the entirely legible Josquin partbook in the hands of the eldest boy demonstrates that the Master was far from lacking the skill to accurately depict musical notation. Instead, there is an evident fascination with text and the textual surface. In the rest of the Master's output, writing and inscription on paper or parchment surfaces were important compositional elements. The 1567 portrait of Katherine de Vere features the golden lettering to the upper left of the sitter, which echoes the intricate loops of the embroidery on her sleeves and bodice; in *Four Gentlemen Playing Primero*, the jack of hearts is carefully rendered and angled to be visible and recognisable to the viewer. The cards betray an intriguing interest in making the invisible visible and playing with the paradigms of the visual art form. Limnings were typically painted on the backs of playing cards. Patricia Fumerton has written evocatively of the tension at play with such a *mise-en-abîme* dynamic:

> [Everything associated with miniature painting] suggests that its habit of public ornamentation kept, rather than told, private 'secrets'. Bedrooms displayed closed decorative cabinets; cabinets exhibited closed ivory boxes; boxes showed off covered or encased miniatures; and, when we finally set eyes on the limning itself, layers of ornamental colours and patterns only show the hiddenness of the heart. As seen in the frequent limning of 'miniatures-within-miniatures', indeed, the regress of concealing layers of ornament extended indefinitely. In Hilliard's *Man against a Background of Flames* for instance, a lover appears to literally bare his burning passion. His fine linen shirt, *en déshabillé,* opens wide to reveal his white breast and an enamelled gold locket hanging from a chain around his neck. Pressed against his heart, the locket undoubtedly contains a miniature of his mistress. (Fumerton 1991)

*Four Gentlemen Playing Primero* and *Four Children Making Music* amplify this tension and the limits of visibility by explicitly depicting where it blurs into what cannot be made visual. The full hand of cards of the two men to the far right and far left of the picture plane are rendered fully visible, their perspective slanted for the benefit of the viewer. In *Four Children Making Music*, the direction of play is inverted. What should be visible-therefore-legible is rendered, in a sense, beyond the scope of what is humanly visible: the closer the viewer gets, the more the shapes of note, stave, clef, and so on unravel and pull apart. There is an evident fascination with the concept of literacy, as well as with the visual as a *read*, decipherable entity.

The sixteenth century witnessed a concerted drive to increase and deepen literacy across England, fuelled in large part by the boom in the printing trade. A huge variety of material became available in the vernacular, from the mathematical to the musical to the anatomical to the poetical; all were designed to reach a readership beyond those who had received a university education. Musical material was no exception and had considerable consequences for the direction that English musical culture ultimately took. In 1562, Sternhold and Hopkins published their psalter, the *Whole Booke of Psalmes* (Sternhold and Hopkins 1562). It would go on to be one of the best-selling books of the century. One hundred and forty three editions of the *Whole Booke of Psalmes* survive from the reign of Elizabeth (Arten 2018, at 151); it is beaten to the most-published spot only by the *Book of Common Prayer*, which reached around five hundred and twenty five editions in English alone by 1729.[10] By the end of the sixteenth century, and due to a flooded market that testifies to its popularity, the *Book of Common Prayer* cost 10d—less than a pound of sugar and just more than a chicken (Swift 2013, at 31). These two books crucially permitted congregations to follow along with the liturgy, allowing them to piece together the words spoken with the visual marks inked on the page and inviting them, slowly but surely,

into a new idiom of literacy. Crucially, included with the prefatory material of the *Whole Booke of Psalmes,* was a brief introduction to fixed-scale solmization, which conceptualised each pitch as having a set, predetermined position on the stave, rather than being entirely mutable (as with the solmization system in use earlier in the sixteenth century). Samantha Arten has noted that this was a decisive moment in the musical culture of the English Renaissance. Sternhold and Hopkins' fixed-scale solmization model slowly, but surely, diffused into the collective consciousness through subsequent editions, throughout the second half of the sixteenth century, such that by the time explicit works of music theory came to be published in the late 1590s, many simply described fixed scale solmization with no reference to the older, mutable system (Arten 2018).

*Four Children Making Music* was painted at the very beginning of the *Whole Booke of Psalmes'* lifetime, in the very culture for which it was first compiled. The children depicted are very likely not part of the democratized intended readership of Sternhold and Hopkins' psalter; they have yet to be identified, but given the other sitters painted by the Master of the Countess of Warwick (including Susan Bertie, Countess of Kent, who educated the poet and musician Aemilia Lanyer, and Dorothy Neville, wife of Thomas Cecil, patron of many English Renaissance composers, and related to the Lady Nevell of *My Lady Nevell's Book*, 1591) (see Note 6), it is likely that they were part of an elite Elizabethan milieu at the heart of the cultural and artistic sphere and received their musical education from a tutor, rather than relying on the *Whole Booke of Psalmes.* Nevertheless, the painting offers tantalising insight into how the textual surface of music notation was perceived in the 1560s, at the time when Sternhold and Hopkins' concept was born, and when the *Whole Booke of Psalmes* first appealed to its readership. The delight in the fundamental slipperiness of legibility and the vagaries of partial literacy is inscribed into the fabric of Figure 1. The forms of the notation and the suggestive non-notation visually echo the finely rendered embroidery on the girl's sleeves, and the coiling quills of the ruffs and cuffs worn by all of the children. They amplify the intricacy of the brushwork needed to capture these elegant details, suggestively eloquent in their vacillation between the figurative and the abstract. Taken together, the impression is of a cohesive visual surface, with a consistent patterning and texture; far from resembling a 'transparent window',[11] it instead approaches representation as an opportunity to revel in the sumptuousness of a surface, such as cloth, or indeed the printed or calligraphed page of a book of musical notation.[12] Given that the painting would likely originally have hung over the textile surface of a wall hanging or tapestry, the desire to render the painted surface as a sort of visual analogue of cloth or fabric is perhaps hardly surprising.

*Four Children Making Music* is a rare surviving panel painting; overwhelmingly, the people of the English Renaissance tended to prefer wall painting, textile, metalwork, and other forms that have a much lower survival rate and are largely associated with the derided 'decorative' arts today, in the twenty-first century. In turning to our next object, a wall painting from 34 High Street, Thame, in Oxfordshire, we can see that the concerns and motivations were similar when it came to exploring the potential of what could be achieved by depicting musical notation in a visual art object. However, what *Four Children Making Music* indicates as to surface is extrapolated microcosmically in the wall painting at Thame, into the textual surfaces implied by print.

## 2. Notational Text

The wall painting at Thame (Figure 3) once adorned the walls of the upper chamber at 34 High Street, Thame, and was transferred with one wall, elements of its ceiling design and both of its adjoining corners almost intact to the Thame Museum in the 1970s. Its schema might appear chaotic to those accustomed to modern approaches towards representation. Multiple scenes, depicting domestic life in the Elizabethan era, clamour for the viewer's attention. One might think that the elements designed to grab the eye first are the textual elements, framed in a scrolled cartouche in the centre of one wall and surviving in fragmentary form in the wall to the right. These word paintings are straight

out of the school of Thomas Tusser's wall posies, but slightly classier in content. The central inscription, in blackletter script, reads:

<div align="center">

The First E

Of Saint Paule for romans

O the depnes of the aboundant

Wisdom of God: howe uncerchable

Are his judgements & his ways paste

Finding out: for who hathe knwen ye

minde of ye Lord? Or who was his con

celoure? Other who hathe given unto

Him first that he might be recompencyd a

gayne? For of him, and through him

And for him are all things

To him be glory for

Ever and ever

Amen

(Hamling and Richardson 2017, at 205).

</div>

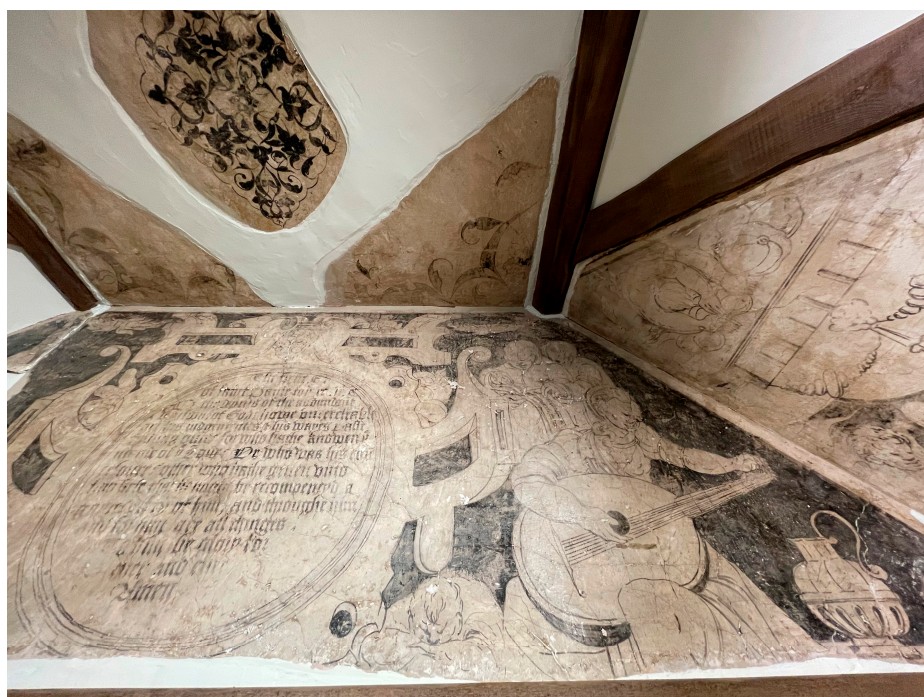

**Figure 3.** Anonymous, wall painting depicting Music-Making from 34 Upper High Street Thame, c. 1560 (Oxfordshire: Thame Museum, photograph author's own).

To the right at the bottom of the wall is another, damaged cartouche, which collapses two sentences from Pythagoras and Aristotle, which had been published by William Baldwin in his *Treatise of Morall Philosophie* (1547). It reads:

Desire nothing of God,

Save what is profitable.

Science is had by diligence:

But discretion and wisdom cometh from God.[13]

Above this second cartouche are cornucopia swags, and the trace of faces gazing up at the text, and across back to the central cartouche and the figures that surround it. To the left of the central text sits a woman playing or tuning a lute, below two children reading from a partbook; echoing them, on the other side, is the fragment of the neck, nuts, pegbox,

and scroll of a bass viol, above another fragment of the edge of its body (the remainder was lost when the wall-painting was uncovered). Their gazes do not cross or interact, but rather direct the eyes of the viewer to yet more elements of the composition. Above them, fragments of moresque/strapwork patterning decorate the ceiling, containing the scene as a whole and providing the sense of an edge.[14]

Amongst the many fragmentary instances of surviving wall paintings from the Elizabethan era, the Thame wall painting is a particularly charming example. Its black and white monochrome is elegantly restrained, reflecting the manner in which the aesthetic of print had seeped into the collective consciousness. The printing press did not simply render books more accessible; the printing trade also produced damask papers (printed papers with decorative floral patterns) to be used as wallpaper (Fleming 2013), or to line the inside of drawers, or to pin up like the modern poster. The monochromatic schema of the Thame wall painting evokes all of these potential usages. Helen Smith notes the dynamic behaviour of what she refers to as the 'writing surface' in the English Renaissance and the way in which the whiteness of the surface becomes the whiteness of writing, of the space in between which undermines the sense of its existence and role simply as writing *surface*: 'writing does not exist 'on' paper, but sinks into the page, in ways that [ . . . ] were experienced by early moderns not only as a practical problem but as a compelling figure for thought' (Smith 2017). In the context of the Thame wall painting, the textualities of surface—and the surface of its textualities—play into a similar dynamic. Tara Hamling and Catherine Richardson note that this chamber, beside the bedchamber on the first floor, was likely a multi-functional space where the family came together to pray, eat, and make music amongst other things, and so that the theme of social harmony is particularly apposite (Ibid., 206). In the interplay of gazes, deictically drawing the eye to other elements within the wall painting as a whole, across and indeed outwards to the room that no longer exists beyond: a life lived through the printed form that provided the family with recipes, prayers, music, decoration, instruction, and entertainment.

This is a textual way of living. I mean this not in the sense that it is bound by words, but rather that it is bound by the *texere*, the woven tissue, of the logic of the material form of print and the experience of literacy as a visual, material process bound to matter. As a technology, it was relatively new. It presented an intoxicating range of potential possibilities and usages: paper could stand in for any number of materials, from flesh to the pure abstract matter of the *phantasia* or imagination.[15] Where it stood in for the matter of music, this operation becomes particularly interesting. Music was not printed in score, but in books of individual parts or in choir books, in which the parts were stacked one on top of each other. The paper form of music thus dramatizes the relationship between performers in a manner that foregrounds their interaction. To read and perform from this kind of music is to be powerfully aware that what you are creating is something collaborative; that you rely upon signs from your fellow performers in order to synergize the performance as a whole. The Thame wall painting alludes to this dynamic. In its interplay of music-making between the woman playing the lute and the two choir boys, across to the lost bass viol and its player (Figure 4). To read from a musical text is not to simply follow a line of music; a performer needed to constantly make eye contact with the rest of the ensemble to feel the beat, to keep together. The brief phenomenon of the table book, as seen in the manuscript collection *A Booke of In Nomines and Solfaing Songes* (1578 and now in the British Library), as well as later editions of Charles Butler's musical-beekeeping treatise *The Feminine Monarchie* of 1609 and John Dowland's *First Booke of Aires* (1597) and *Lachrimae, or Seaven Teares* (printed in 1605), testified to the importance of this way of musical reading with parts arranged in the round, with the book intended to be placed in the centre of a table (Dowland 1609). The mise-en-page of table books did not catch on, but this is likely due to the fixed positioning of parts, rather than the 'in the round' aspect.

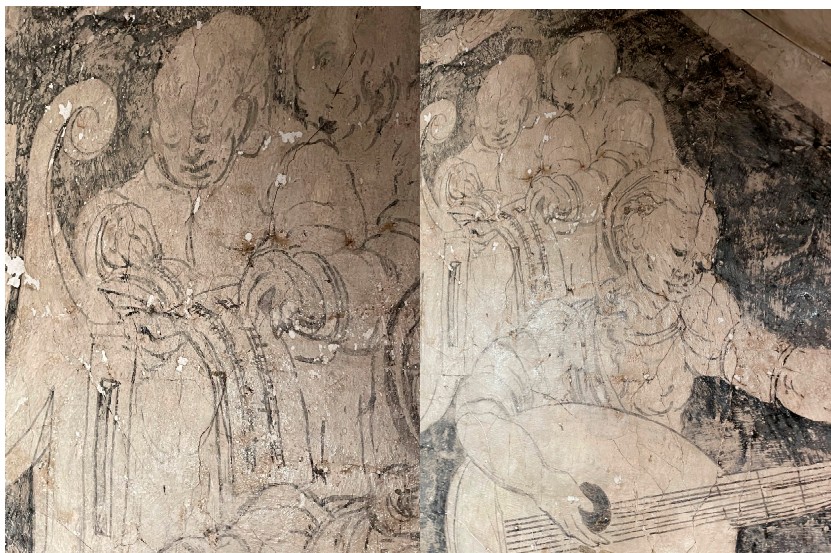

**Figure 4.** Detail of singing children and woman playing the lute, from the wallpainting from 34 Upper High Street, Thame.

There is another aspect of the physical form of music books that percolated into the logic of the Thame wall painting. The pattern fragments from the ceiling (Figure 5) suggest that the design might once have filled the whole room, and that those areas that were not treated to figurative elements were decorated in a strapwork or arabesque scheme to be found in the margins and at the beginning and ends of individual pieces in some of the music books of the period. Now known as printers' flowers or printers' lace, they were a vital part of printing technology in general in the English Renaissance. They supported non-textual areas of the page, ensuring that more explicitly semantic elements did not smudge or slip during the printing process. They could function much in the same way as maker's marks. And, in an age of widespread partial literacy, they served as a guide for the eyes, thematizing the undulations of words and echoing their forms in an almost asemantic manner. Their passing resemblance to flowers, notes Jessica Rosenberg, was not the important aspect:

> The printer's flowers that frame titles and that mark off division within volumes are not, of course, *actually* flowers. The winding patterns that fill these inked ornaments, sometimes known as 'vinets', fall under the category of ornament that historians of art and design call 'arabesques'—elaborate winding designs that originated in Islamic art and design and first appeared in printed books in sixteenth-century Venice, before being developed and copied by French and then English printers. Though some were based on botanical forms, their intricate patterns are abstracted from worldly referents. [ . . . ] Instead, they draw the eye to the patterned surface of the page, where they seem to work *like* letters—filling up space and taking form—without being legible *as* letters. (Rosenberg 2022)

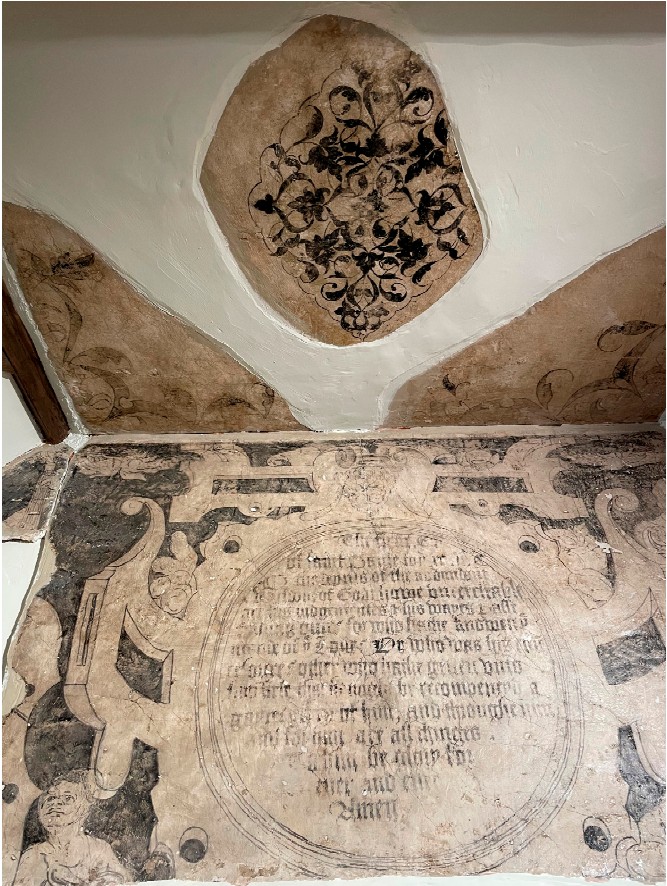

**Figure 5.** Detail of ornamental ceiling schema and cartouche from the wallpainting from 34 Upper High Street, Thame.

They were, instead, a way of marking habitation: as delineating a space as for a particular use generally and (in the case of the Thame wall-painting) as a space for music-making. Looking at the wall painting conjures a sense of enclosure. It is a celebration of the physical form of the printed music book, just as it was becoming a celebrated and desired mode of materiality through books, such as the *Whole Booke of Psalmes.*

The question of whether or not the notation depicted in the Thame wall painting is legible is, fundamentally, irrelevant to the way that it generates meaning. The painting is a celebration of the concept of the printed book and of literacy. It is of use to musicological scholarship and investigations into the history of music because through its allusions to the visual appearance of the printed book, it provides a new way for thinking about notation: as evidence of the ways those first readers of English printed books took aesthetic pleasure from the monochromatic patterns of notation. Fumerton notes that the blackletter typeface used in broadside ballads—similarly made up of ornate print blocks that could be read, but also which were delightful to the eye in and of their own right—increasingly served visually as an art form for the partially literate.[16] It was extremely common, across the social spectrum, to paste blackletter ballads on walls the way one would a panel painting, painted cloth or applique hanging. Such thinking extends easily to the twisty, patterning forms of musical notation, so similar to the strapwork forms of decoration that filled the textiles, woodwork, and metalwork of the Elizabethan era. Likewise, as Christopher Page notes, it is highly unlikely that the Thame wall painting was a rare depiction of music-making (Fleming and Page 2021); the relative lack of other surviving examples can largely be attributed to the craze for rebuilding across the seventeenth and eighteenth centuries, as demonstrated by the fact that, like other surviving depictions of music-making such as the frieze from the Great Chamber at Gilling Castle, the grand staircase from Slaugham

Place, West Sussex, and the wall painting from Park Farm, Hilton, the Thame, wall painting survived because it was *removed* from its architectural context. The fortunate partial survival of the Thame wall-painting offers a tantalising insight into a way of thinking about musical notation that allowed it a value free from whether or not it communicated a coherent or performable piece of music.

Illegibility, within the matrix of the Thame wall painting, becomes a key element of its visual schema. The three staff lines of the partbook (Figure 5) held by the singing children (as opposed to the five staff lines typically in use in notation during this period) do not prevent interpretation as music: this is not their chief impact upon the effect of the wall painting as a whole. Rather, they offer a multitude of different ways to interpret the music. They are hyper-legible. Each staff line and space could connote a multiplicity of pitches in a manner that recalls the fluidity of hexachordal solmization, and the culture of musical reading that England had inherited from centuries of theoretical convention. Rather than frustrating its original viewers with its lack of illusionistic faithful depiction of musical notation, this would have been a source of visual delight. The representational conventions of the English Renaissance, inflecting the way in which people expected to be able to encounter and interpret a visual art work, meant that the ambiguity of the music notation would have served as an additional enticement to its original viewers. It was a virtue, rather than an artistic failing. Coupled with its playful allusion to the aesthetic of the printed prayer book, the Thame wall painting offers us a beautiful insight into what musical reading once meant to its original viewers.

### 3. Conclusions: Notational Texture(s)

This article set out to problematize the association of sign and signified when it comes to representations of il/legible musical notation in the visual art objects of the English Renaissance. Through the lens of *Four Children Making Music* and the Thame wall painting, two roughly contemporaneous paintings from across the social spectrum, it has demonstrated the variety of ways in which illegibility served as a virtue and as a means to invite the viewer to intervene within the fabric of the image. In both, questions of il/legibility offer us a snapshot into multiple facets of the period: the printing markets, ideas surrounding literacy, and of course music-making. All three are united by the way in which their inclusion of music books snags the eyes and creates visual interest. In our concluding case study, we shall see that this is broadly what motivates the inclusion of musical notation that are partially legible, or with pretensions to legibility.

Our final example is a furnishing panel from a series of eight depicting the Liberal Arts, made in the second half of the sixteenth century for Bess of Hardwick, Countess of Shrewsbury, at new Hardwick Hall. *Grammar*, *Logic*, *Rhetoric*, *Arithmetic*, *Astrologie*, *Perspective*, and *Architecture* (in place of Geometry) and *Musiques,* are each picked out in sumptuous embroidered and appliqued taffeta and linen panels on black velvet (Levey 2010). Each figure, decked out in pseudo-classical dress, is constructed on a linen base with the flesh parts covered in plain silk and then embroidered in stem and back stitch and delicately shaded in paint to pick out the details of facial features and so on. *Musiques* (Figure 6), ripped down the middle, has been lovingly stitched onto a linen ground to keep the two sides together. She is in no worse state than her sister arts, and indeed the fact that she is still framed by her full classical arch (rendered in striking cloth of gold with inserts of blue velvet, now faded to green, and embellished with scrolling giltwork embroidery) makes her one of the three in best condition. The collection is rather unusual; the substitution of *Perspective* and *Architecture* in place of Geometry and the minute care in rendering the instruments of each Art, down to the last tiny stitch and shading, are unmatched in any other surviving example of English textile work of this period[17] Even amongst such remarkable company, something remarkable is at play in *Musiques*: in her hands, where convention might imply that she would be holding a musical instrument to represent her 'Art', she loosely clutches a book. At her feet, we see what might be the same book lying open (Figure 7). Its tiny silk pages are inked with miniscule musical notation.

Closer inspection reveals that it is almost legible—a C4 (perhaps a C3 alto) clef, a phrase of minims and semi-minims (now crotchets, in modern western musical parlance), spreading at a jaunty angle across both pages.

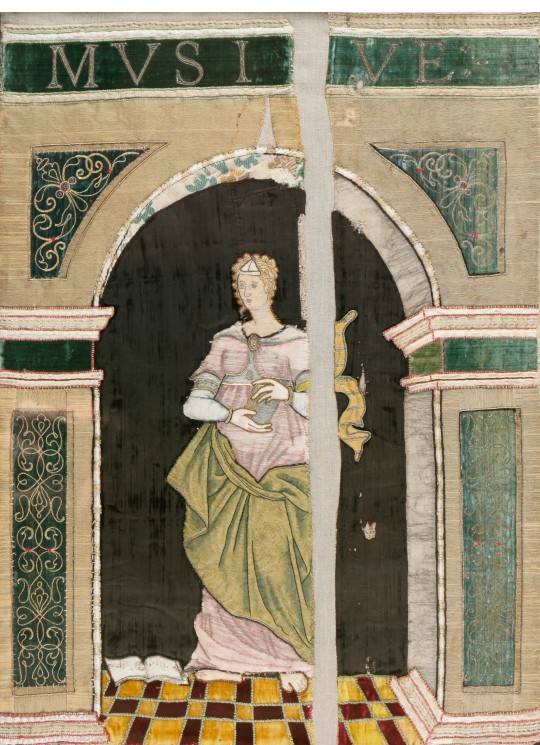

**Figure 6.** Anonymous, 'MVSIQVES' furnishing panel depicting the figure of Music with a music book, appliqué silk and velvet scraps with embroidery in couched gilt and silver-gilt thread and spangles in candelabrum and scrolling stem patterns over a linen base, second half of the sixteenth-century. National Trust: Hardwick Hall, Derbyshire. Object No. 1129552.

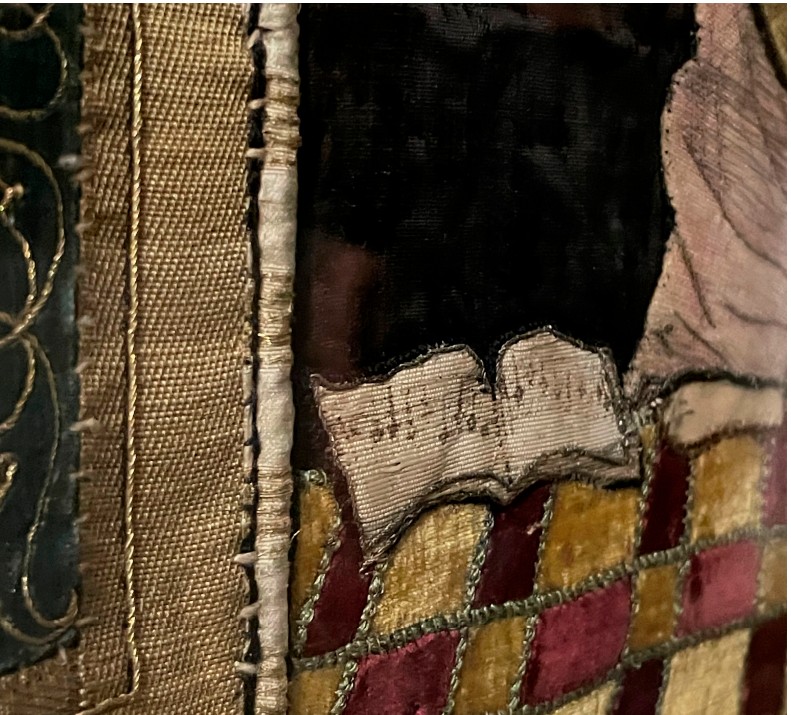

**Figure 7.** Detail of the 'MVSIQVES' furnishing panel, showing the musical notation (photograph author's own).

The *Musiques* furnishing panel is a little different to our other two examples: we can read the music it depicts, and we can make out the beginning of a phrase. Nevertheless, the way that it seeks to snag the eyes of its viewer speaks to the same impulse and culture of visual expectation as *Four Children Making Music* and the Thame wall painting. In order to make out the notation, it is necessary to lean in very close to the furnishing panel. At a typical distance, it is merely a sumptuous pattern, such as the damask of the pillars, a texture similar to the once-blue velvet. Due to the fact that the panel has been cut down and repeatedly repaired, we do not know for certain what its full material form once was; it is possible that it once upholstered a series of chairs or cushions. What is clear, however, is that the furnishing panel was designed to be viewed in a highly intimate setting, as you sat against that chair or cushion, perhaps even as you listened to a performance. It rewards repeated viewing, and multiple, rather than fixed, perspectives; those tiny, inked marks of notation entice you to move around the object to appreciate its full scope. In scale it differs vastly from *Four Children Making Music* and the Thame wall painting, but each of these objects invite the viewer to adopt a synoptic way of seeing, all of which impinges on the *trompe-l'œil* of the music notation that features in each.

English Renaissance art has historically been condemned for its inability (or, more accurately, its refusal) to adopt the illusionistic register of its Florentine contemporaries. Through the lens of illegible music notation, this article has demonstrated that a very different model of realism and representation is at play in the art objects of this period. Entertaining illegibility or insouciant or partial pretensions to legibility as a deliberate visual strategy the *Musiques* furnishing panel, *Four Children Making Music*, and the Thame wall painting reveal a particularly English approach to depth and texture as a model of visual engagement. It alludes to the contemporary taste for printers' flowers and blackletter, both forms that evoke the appearance of text (written and printed) without its semantic import. Ultimately, it advocates for the value that such instances offer for those interested in the musical culture of the English Renaissance. All three case studies were created at a crucial turning point in the way that musical literacy was approached and understood; all three reveal that these developments were a source of fascination and delight. They were not created with the intention of providing an illusionistic record of what music was performed where, and they were certainly not created for the benefit of scholars approaching them as sources of evidence today, in the twenty-first century.[18] Depicting a musical text as illegible in the English Renaissance should not suggest that their makers were unaware of what notation looked like, but, rather, implies that they were interested in the complications and opportunities brought about by the gradual blossoming of musical literacy. The manner in which each of these visual art objects explored these complications and opportunities reveals a range of representational strategies, which are alien to our eyes today, but which speak to a once lively musical-visual culture.

**Funding:** This research was funded by Leverhulme Trust Grant No. ECF-2020-071.

**Conflicts of Interest:** The author declares no conflict of interest.

## Notes

1.   Kerry McCarthy, 453–54.
2.   Tim Shephard and Sanna Raninen 'Music, notation, and embodiment in early sixteenth-century Italian pictures' in (Schuiling and Payne 2022), at 80.
3.   For more on this see Tara Hamling, 'Living with the Bible in post-reformation England' in (Doran et al. 2014) at 212.
4.   See for example Lucy Gent's 'way of seeing': 'A way of seeing that takes cognizance of the Elizabethan obsession with mutability as reflected in so many appurtances—cushions, leaden spoons, furnishings, 'twists of rotten silk', paints, fabrics, books—all the increasing impedimenta of a rising standard of living sitting around in our damp climate. It would also be a sight virtually implicated in the materials of what it sees: the grainy textures of old brick, and of embroidery whose close stitches offer us a picture without any suggestion of looking through a transparent window'. Lucy Gent, 'The Rash Gaze: Economies of Vision in Britain, 1550–1660', in (Gent 1995).
5.   Katherine Coombs and Alan Derbyshire, 'Nicholas Hilliard's Workshop Practice Reconsidered', in (Cooper et al. 2015, at 246–47).

6  Kerry McCarthy, 'Josquin in England', 450.

7  (Strong 1983). For more on Teerlinc, see (Saltmarsh 2020).

8  As Town notes: 'These portraits share the same static posture of Bettes and Holbein but do not convey the stillness or serenity of their work. The influence of Eworth can also be seen in the clasped hands and the verse and prose inscriptions extolling the virtues of his sitters, but the limitations of his draughtsmanship result in the portraits failing to deliver the same pious intensity of a portrait such as Eworth's *Elizabeth Roydon* of 1563'. Edward Town (2020).

9  Edward Town, 'A Portrait of the Miniaturist as a Young Man'.

10  For a discussion of the life and development of some of the earlier editions, see (Blaney 2021).

11  Lucy Gent, 'The Rash Gaze', 379.

12  My thanks to Liz Oakley-Brown, Kevin Killeen and the participants of the 'Scrutinizing Surfaces' conference at Lancaster in 2015, for galvanizing my thoughts on the early modern surface; see especially (Canavan 2017).

13  Ibid; for more on the use of aphorisms in domestic contexts, see (Hamling 2014, at 221–27).

14  Katherine Butler, 'Printed borders for sixteenth century music or music paper and the career of music printer Thomas East' in *The Library*, XIX/2 (June 2018), 174–75 and 205. See also (Fleming 2006, 2016).

15  Ibid; see also (Reynolds and Bellingradt 2021; Ferrell 2008, 2010).

16  (Fumerton 2020); email correspondence with the author, September 2022.

17  For more on the panels, and the decoration of Hardwick Hall in general, see (Wells-Cole 1997).

18  For more on this in the context of Italian art, see (Shephard and Raninen 2022).

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
