# Peer review of "{Not}ation: The In/Visible Visual Cultures of Musical Legibility in the English Renaissance"

_arts, 2023_

Round 1

Reviewer 1 Report

I very much enjoyed this fascinating analysis of the significance of musical notation in specific examples of sixteenth-century English art. The article draws attention to generally neglected details within well and lesser known artworks to reflect on questions of legibility and creativity. Close engagement with the detail of the first painting allows the author to draw us in with questions prompted by the ‘enigmatic and suggestive imitation’ of musical notation in the second of the open books. The article argues persuasively that the choice to keep this notation illegible was deliberate, and with the further examples of the Thame wall painting and the Hardwick Liberal Arts embroideries (depiction of Music), prompts a re-evaluation of musical notations in visual culture of this period. It forces a move away from standard art-historical expectations of realism, and associated value judgements when such realism is lacking, to a more authentic early modern way of seeing and reading; a negotiation between the legible and illegible. It successfully argues that illegibility could be a source of visual delight, that ambiguity was an enticement to creativity in responding to art works, allowing endless musical [and other cultural] possibilities. 

I support the publication without the need for corrections to the main text (apart from the missing footnotes early on and the few typos listed below). I do, however, offer a few suggestions that could enhance the piece further should the author wish to incorporate them. 

The abstract is framed more as a set of questions than argument, which is fine but I think you might want to offer the reader a bit more here. The final line could be extended to state your over-arching argument more explicitly. I'm only suggesting one further sentence.

Line 28: The reference to the finding by Kerry McCarthy needs a footnote.

Line 34: 'has rightfully been interpreted as a prime insight into the musical and visual cultures of the mid-sixteenth century'. – needs footnote

Line 59: This sentence seems rather too bald: 'The English Reformation, enforced from above rather that popularly driven, a protracted drift away rather than a sudden break, resulted in a distinctive visual culture.' What timescales are you thinking of here? At what point did it cease to be a 'top down' movement? Can you develop or nuance this?

Line 43: Explain: ‘mul-ti-layered materiality’, or is this part of a more general issue with several lines/words containing unneccesary hyphens in this section?

Line 59: typo that / than

Line 62: mark – marker? And how distinctive to England was this trend?

Line 159: query if word 'other' was in the original quotation as it seems unnecessary.

Line 390: reference to Thame painting as isolated incident of music making – there are others (though not painted directly on wall, e.g. Gilling Castle frieze).

Line 437: suggest rephrase' the substitution of Geometry for Perspective and Architecture' to 'the substation of Perspective and Architecture in place of Geometry' (otherwise may suggest substitution the other way around).

Especially for the discussion of the Hardwick Liberal Arts, but also relevant more generally is extent of faithfulness in copying from specific printed sources – how far is the musical notation accurately reproduced from the print? Suggest referencing Antony Wells-Cole's book Art and Decoration (YUP, 1997) here.

Lines 479/480: missing word: were unaware of what notation looked [like]?

Line 481: missing character - Brough[t]

The author might also find this article on painted inscriptions in wall painting of interest: Tara Hamling, (2014). Living with the Bible in post-reformation England. In J. Doran, C. Methuen, & A. Walsham (Eds.), Religion and the Household (pp. 210-239). The Boydell Press.. 

Author Response

Thank you for taking the time to review the article, and for your insights and pointers! I've adjusted the article to your recommendations, and have added footnotes and given the text a thorough copyedit.

Specifically, I have added nuance to line 59; as the reviewer rightly notes the situation had changed significantly by the mid-sixteenth century.

With regards to line 43, this seems to have been an issue with the formatting of the text in the MDPI template (it - and other extra-hyphenated instances - are not in my original document). I'll ensure these are eradicated from the revised text!

With regards to line 159, the extra 'other' is indeed in the original text.

I read Hamling's article with interest! Thank you for bringing it to my attention.

Reviewer 2 Report

This is an inventive and sensitive analysis of three English Renaissance images featuring music-books. The author is able to call upon a wide, cross-disciplinary bibliography, and to find points of contact between musical notation and wider theories of writing, reading and looking in the period. The argument feels at times quite theoretical or intuitive, with the result that the author’s conclusions land as provisional rather than definitive, but they are insightful and thought-provoking nonetheless. There are a few places where the argumentation needs sharpening up a bit, indicated below. Once these small revisions are dealt with, I strongly support publication.

p3: “This six-note solmization system existed alongside the seven-note concept ‘A’, ‘B’, ‘C’ and so on of the diatonic system, and the as-yet unelucidated theory of the eight-note pitch class we now call octaves.” – it isn’t accurate to say that the theory of the 8ve scale was as-yet unelucidated: modal theory already described 8ve scales.

p8 I think the differences between Renaissance and modern musical textuality and reading are overplayed here. “Music was not printed in score, but in books of individual parts or in choir books, in which the parts were stacked one on top of each other. The paper form of music thus dramatizes the relationship between performers in a manner that is not evident in modern sheet music.” – well, most modern notated ensemble music is also performed from, essentially, partbooks; it’s only in choral music that it’s common to perform directly from a score. “To read from a musical text in the English Renaissance was not to simply follow a line of music; a performer needed to constantly make eye contact with the rest of the ensemble to feel the beat, to keep together.” – yes, but this remains the case in ensemble performance now, whereas you seem to be presenting it as a point of difference from modern practice.

p9 “The pattern fragments from the ceiling (Fig. 4) suggest that the design might once have filled the whole room, and that those areas that were not treated to figurative elements were decorated in a strapwork or arabesque scheme to be found in the margins and at the beginning and ends of individual pieces in music books.” – this is a little misleading; for sure, such elements can be found in SOME music books, but there are also some that are pretty plain. 

p9 “It is of use to musicological scholarship, and investigations into the history of music because it communicates just how much aesthetic joy and pleasure those first readers of English printed music books experienced from the monochromatic patterns of notation.” – I’m not quite clear, from the aforegoing arguments, how it communicates aesthetic joy and pleasure.

p10 "The three staff lines of the partbook (Fig. 5) held by the singing children (as opposed to the five staff lines in modern western classical notation)” – 5 lines was also conventional in 16th-c notation, except in plainchant, so the difference you’re implying here from modern practice is somewhat false. It is true that 16th-c practice allowed more flexibility in staff arrangement than modern practice, but the vast majority of the time it was 5 lines.

Author Response

Thank you for taking the time to review the article, and for your insights and pointers! I've adjusted the article to your recommendations, removing the inferences on modern music practice (relics of the most recent time I presented this research, to a non-music specialist audience), as noted on p8 and p10 with regards to staff lines and wherever I've found throughout the text.

On p. 9 I've clarified my statement; my point was to encourage musicological interest beyond the notes on the page towards the general, meta form of the music book.

With regards to p.9 on Fig. 4 I've tidied up the muddiness of expression, and clarified that printers flowers appear in some music books (and that this is about the aesthetic of the book in general); likewise on p. 3 I've removed references to the 'unelucidated' theory of the octave.